# Microcamera Visualisation System to Overcome Specular Reflections for Tissue Imaging

**DOI:** 10.3390/mi14051062

**Published:** 2023-05-17

**Authors:** Lorenzo Niemitz, Stefan D. van der Stel, Simon Sorensen, Walter Messina, Sanathana Konugolu Venkata Sekar, Henricus J. C. M. Sterenborg, Stefan Andersson-Engels, Theo J. M. Ruers, Ray Burke

**Affiliations:** 1Biophotonics @ Tyndall, IPIC, Tyndall National Institute, University College Cork, T12 R5CP Cork, Ireland; 2Netherlands Cancer Institute–Antoni van Leeuwenhoek, 1066 CX Amsterdam, The Netherlandsh.j.sterenborg@amsterdamumc.nl (H.J.C.M.S.);; 3Group Nanobiophysics, Faculty TNW, Twente University, 7522 NB Enschede, The Netherlands; 4BioPixS Ltd.–Biophotonics Standards, IPIC, Lee Maltings Complex, Dyke Parade, T12 R5CP Cork, Ireland; 5Department of Physics, University College Cork, T12 K8AF Cork, Ireland

**Keywords:** micro cameras, imaging, specular reflection, surgical guidance, endoscopy, multi-flash, cross polarisation

## Abstract

In vivo tissue imaging is an essential tool for medical diagnosis, surgical guidance, and treatment. However, specular reflections caused by glossy tissue surfaces can significantly degrade image quality and hinder the accuracy of imaging systems. In this work, we further the miniaturisation of specular reflection reduction techniques using micro cameras, which have the potential to act as intra-operative supportive tools for clinicians. In order to remove these specular reflections, two small form factor camera probes, handheld at 10 mm footprint and miniaturisable to 2.3 mm, are developed using different modalities, with line-of-sight to further miniaturisation. (1) The sample is illuminated via multi-flash technique from four different positions, causing a shift in reflections which are then filtered out in a post-processing image reconstruction step. (2) The cross-polarisation technique integrates orthogonal polarisers onto the tip of the illumination fibres and camera, respectively, to filter out the polarisation maintaining reflections. These form part of a portable imaging system that is capable of rapid image acquisition using different illumination wavelengths, and employs techniques that lend themselves well to further footprint reduction. We demonstrate the efficacy of the proposed system with validating experiments on tissue-mimicking phantoms with high surface reflection, as well as on excised human breast tissue. We show that both methods can provide clear and detailed images of tissue structures along with the effective removal of distortion or artefacts caused by specular reflections. Our results suggest that the proposed system can improve the image quality of miniature in vivo tissue imaging systems and reveal underlying feature information at depth, for both human and machine observers, leading to better diagnosis and treatment outcomes.

## 1. Introduction

In a clinical setting, bio-photonic techniques play a role as complementary methods which can increase the breadth of diagnostic options, with quick turnaround spot testing that can aid clinical decisions. Gathering data for evidence-informed decisions, in particular in a fast-paced medical environment, can be in competition with acquisition time and the volume of potential data available. Therefore, evidence should be rapidly available when and where it is needed and be relevant to the task performed [1,2]. In vivo diagnostic imaging methods such as diffuse optical imaging [3,4], diffuse reflectance spectroscopy [5,6,7,8], fluorescence imaging [6,9], optical coherence tomography [10] and in vivo microscopy [11] are undergoing continual development as medical diagnostic techniques. Applications range across medical disciplines, from screening, to tumour margin detection and surgical guidance, and many techniques have been adapted into medical devices seeing real-world use. In order to fit into the clinical workflow, these technologies are now being integrated into handheld or endoscope-based imaging probes, paving the way for their introduction into clinical settings. Such integrated sensorised probes are usually either fibre- [12,13,14] or camera-based [15,16]. For example, camera-based hyperspectral imaging probe, as well as camera-based surgical pen type fluorescence probes for diagnosing tumour status, are actively being researched. Currently, different imaging probes are being investigated for usability in tumor imaging [17] and fluorescence-guided surgery [5]. Breast cancer monitoring is an example where handheld probes have suggested use cases [18]. The development of sub-mm sized micro cameras, such as the AMS NanEye device, offers a promising platform for small-footprint medical imaging and work to develop applications is ongoing [19,20]. The main advantage of placing a camera on the tip of an endoscopic probe is in providing flexibility for the user. Such a device can aid and complement decision making by potentially acting as the sensor while also providing a guidance image.

Due to variations in camera-light sensitivity and frequent operation in dark environments in the body, medical imaging probes require external illumination. The ability to deliver various wavelengths to the target, for targeting specific bio-markers, controlling penetration depth, and multi- and hyper-spectral imaging applications are of interest for tissue imaging. Illumination is often provided by means of distally mounted LEDs, or in the case of this paper using optical fibres. The use of a point source for illumination causes specular reflections. These reflections are direct reflections from the surface and preserve the polarisation of the illumination source. The glossy and smooth tissue surface and clinical lighting environment give rise to large specular reflections, which can obstruct vision particularly in the most popular feature analysis domain, that of colour [21].

Because of the necessity to provide clear data to clinicians and image-processing algorithms, the image should be free from undesirable artefacts. Additionally, a pixel saturated by specular reflection obscures the information available in that area, and underlying features that may be of interest become unavailable for viewing. Image-processing algorithms in particular may produce errors in algorithms that segment and detect shapes [22]. Therefore, minimising or eliminating the effects of specular reflections should be a priority for image-based probes. There exist a number of techniques to remove specular reflections [23], that are either software or hardware based. One such multiple-image-based approach uses multiple illumination points to shift the specular reflections in subsequent images of a scene. This allows for a mixed software- and hardware-based approach which eliminates overlapping and partially overlapping reflections [24,25]. This multi-flash approach collects more light and does not require expensive filters. A purely hardware-based approach employs linear polarisers in an orthogonal, or crossed, arrangement, to remove specular reflections. This is a well-known technique that has seen wide commercial adoption in professional photography and eye wear. It has also seen use in the removal of corrupting glare from hyperspectral images [26] and been shown to have application in medical tissue images [4,12]. The deterioration of polarisation state occurs rapidly in tissue due to multi-scattering events. Orthogonal polarisation imaging exploits the fact that direct reflections as well as single scattering events maintain polarisation [27]. However, as light propagates through tissue and enough scattering events take place the polarisation state is subsequently lost. The rate of loss is dependent on the polarisation state of the incident light. The background theory and further details on de-polarisation of light in tissue is well described in the literature [28,29,30,31,32]. By linearly polarising the incoming light, while detecting with an orthogonal polariser, the diffusely reflected component of the light can effectively be filtered.

The objective of this work is to study these aspects of diagnostic potential to guide developments that can provide the clinician with tools for rapid imaging and decision making, with line of sight to diagnostics, by developing a microcamera-based platform that is capable of overcoming the challenges of specular reflections. There has not been any clear study to determine which of these two techniques is better for tissue imaging to reduce specular reflection. The importance of specular reflection removal to fully image subsurface structures in optical diagnostic imaging, and the depth dependence of this imaging has not been studied. Furthermore, miniaturisation of both techniques to this scale has not been demonstrated previously. In this work, we present miniaturised multi-flash and cross-polarised systems using micro cameras. They integrate well into clinically relevant handheld probes, while allowing for miniaturisation to 2.3 mm endoscopic probes which can navigate into peripheral spaces in the body.

The scope for miniaturisation is desirable, as well as the possibility to extend device capability into other domains aside from traditional imaging, such as sensing. This refers to the delivery of multiple wavelengths and fibre connectivity to enable multi-modality. In this paper, the groundwork for a clinical imaging system is presented. Two imaging probes with fundamentally different specular reflection removal techniques are presented in the context of clinical tissue images. Images of phantoms and tissue are acquired, and both specular-reflection removal methods are demonstrated to effectively preserve information from pixels that have been saturated by specular reflection. An analysis of imaging depth vs. illumination wavelength for different techniques is performed. A light source capable of custom variable illumination is used for illumination and a control code ensures rapid data acquisition. The ability to image using microcameras through diced polarisers is demonstrated.

## 2. Materials and Methods

### 2.1. Multi-Flash Illumination Micro-Imager Probe

A schematic of the experimental imaging probe is shown in Figure 1. A 1.0 × 1.0 mm^2^ AMS NanEYE RGB micro-camera, with a field-of-view (FOV) of 90° and F# 2.7, is integrated into a 3D-printed handheld imaging probe. Additionally, four 0.5 NA, 400 μm core diameter illumination fibres (Thorlabs FP400ERT) are positioned concentrically around the camera, at a distance of 2.4 mm, to provide uniform illumination with the intention of maximising the illuminated FOV of the camera sensor from each fibre position. The handheld probes are designed for a larger diameter of 10 mm for simple handling during the experimental phase. This arrangement also allows for further miniaturisation, which is discussed later. The illumination fibres are separated to ensure overlap between the emitted light cones, both for maximising illumination power in the case of all four fibres being used for illumination and ensuring that the multi-flash algorithm functions correctly by shifting the light cone with subsequent flashes.

The optical fibres allow light to be delivered to the sample from four slightly offset illumination locations. A shift in specular surface reflections can thereby be induced which can later be low-pass filtered during post-processing eliminating the more intense surface reflections. A one-to-four optical switch (Leoni mol 1 × 4) allows for quick switching between illumination fibres, minimising effects of movement by allowing for acquisition times of under half a second limited by the synchronisation with the frame grabber. The system is controlled using a LabView (National Instruments, Austin, Texas) based control software. Multiflash images were analysed using a multiflash algorithm [24,25] and reconstructed using a MATLAB v2021a (Mathworks, Natick, MA, USA) code and Poisson image reconstruction based on a sine transform [33,34,35]. The small shift in the location of the specular reflections was used to eliminate the reflections from the image. The choice of fibre to illuminate allows additional flexibility in illumination wavelength selection.

### 2.2. Orthogonal-Polarisation Micro-Imager Probe

The cross-polarised version of the probe incorporates the same physical design as the multi-flash probe. High-quality, glass substrate, linear polarisers, selected for high transmission and contrast across the visible and near-infrared (NIR), with average transmission in the visible of 90% and up and a contrast ratio of 1500 at 650 nm (MoxTek RCV8N2EC) were mechanically diced to 2.0 mm square and mounted in front of the camera sensor and the optical fibres. The orientation of the polarisers was such that orthogonal polarisation was achieved between the camera and each illuminating fibre. The polarisers were cut using a mechanical silicon wafer dicing machine, and their size was chosen to be the maximal permissible size given the desired probe dimensions. The dicing saw was used to partially cut through the glass, and the small components could then be separated from the bulk by carefully cracking them off. We have since achieved a dice of 0.50 × 0.50 mm enabling further miniaturisation.

The experimental set up along with a close-up view of the tip of the probe, showing the UV-glued polarisers are shown in Figure 2 above. In this probe, illumination is delivered through each of the four fibres simultaneously. Therefore, the optical switch is not required when utilising this set up. The use of four fibres simultaneously also allowed for a good illumination profile, which, given adequate light intensity and coupling on the back-end, overcomes some of the losses due to the polarisers.

### 2.3. Light Source and Portable Set Up

Illumination was delivered using a custom-built, portable, fibre-coupled multi-LED light source, designed and built in house. This work will be published separately. The source is capable of illuminating with five different wavelengths from 400 nm to 940 nm. Wavelengths were selected to allow for the reconstruction of white light using a combination of red (660 nm—Thorlabs M660D2), green (540 nm—Thorlabs M530D3), and blue (450 nm—Thorlabs M450D3) wavelengths. Moreover, two NIR wavelengths were selected to provide deeper penetration depth in tissue. These were at 850 nm (Thorlabs M850D2) and 940 nm (Thorlabs M940D2). The LEDs are coupled into a single optical fibre using an arrangement of dichroic mirrors. This results in a fibre-coupled source which can be used plug and play style with either of the two aforementioned measurement configurations. The wavelengths choices are adaptable and can be adjusted depending on application.

The entire apparatus is mounted on a movable trolley as shown in Figure 3. This allows the system to be portable and easily transported between the laboratory and operating theatre. The probes can be swapped by simply plugging them in to the LED source. The probes themselves can be used freehand or mounted in a fixed position above a sample. The laptop allows for near real-time data acquisition and control, with the camera data being parsed through a custom-made readout board (BAP Image Systems, Ergolding, Germany) which interfaces with LabView.

This portable trolley set up provides a portable system. The probe can be used handheld when desired, and a system such as this has a form factor which is acceptable in a clinical setting. When capturing images for validation the sample is placed in the sample holder and the probe fixed at a set distance of 8.0 mm above the sample to ensure the field of view remains fixed for consecutive acquisitions.

### 2.4. Ex Vivo Validation

#### Tissue Mimicking Phantoms

The primary goal was to verify the specular-reflection-removal techniques on a highly reflecting standard. Tissue-mimicking phantoms (BioPixS0020 by BioPixS, Cork, Ireland), with well-defined scattering and absorption values to match the optical properties of human tissue, were used to verify functionality of both probes. The phantoms, seen in Figure 4 consist of a base phantom and a number of top layer phantoms. These can be combined together to create range of multilayer phantoms with different top layer thickness while the air gap in eliminated due to the phantom design to avoid optical boundary effects. Well defined layers of depth 0.50 mm, 1.0 mm, and 2.0 mm, allowed also for a verification of feature detection at different imaging depths with both probes. This is of particular interest when using near infrared illumination which has a deeper penetration depth. The phantoms present a glossy surface and contain embedded features of hydroxyapatite of 1 mm diameter. The phantoms therefore allow a deliberate challenging of specular reflections, presenting even more significant surface reflections than what can be expected from real biological tissue. They allow for repeatability and unlike tissue samples do not dry out. The hydroxyapatite particles were used to mimic a feature of interest to a clinician, who may be looking to detect abnormalities embedded in tissue. Hydroxyapatite is similar to human hard tissue and a suitable analogue to mimic the imaging of cardiovascular plaque, for example, while the phantom is not directly analogous to a human organ and imaging phantoms in a controlled environment does not translate directly to live observation in a human or animal, the phantom does provide optical properties that mimic the light transport through the tissue. Therefore, it is suitable when investigating the ability to resolve sub-surface features after specular reflection removal.

### 2.5. Ex Vivo Human Tissue Imaging

To validate this system on tissue, images of breast slices after breast conserving surgery were acquired. The glossy surface of breast specimens were used to test the capacities of the system on human tissue. Images of excised human breast tissue were acquired using the aforementioned probes as part of a post-operative pathological workflow. An image of the clinical set up used to image the excised tissue samples is shown later in the text. The tissue samples were imaged using both the multi-flash capable probe as well as the polariser probe. Regions of interest on the tissue samples were selected, and subsequently imaged.

## 3. Results

### 3.1. Ex Vivo Validation: Phantom Images

The performance of the multi-flash algorithm as well as cross-polarisation imaging was validated in the laboratory before introduction of the devices into a clinical settings. Images of phantoms captured using both probes are presented in this section using different illumination wavelengths and phantom depths. The reconstruction algorithm for the multi-flash method was also validated.

Figure 5 shows an example of the multi-flash system’s image outputs. Each illumination fibre sequentially illuminates the sample, in this case a bare tissue phantom. The four images that are generated can be seen in Figure 5a–d where it is clear that the illumination pattern changes as the scene is illuminated from slightly different positions. The specular reflections present themselves as very bright areas in the center of the illuminated area, where the detector saturates and the hydroxyapetite features are not discernible. The specular reflection shift can also be seen in each of the four images where the saturated region is observed in a different location of the sample. Figure 5e shows the reconstructed output of those four images. The bright saturation due to direct surface reflections is filtered out and the margins of the features are visible once again. The reconstruction error of this method was examined by feeding four identical images into the reconstruction algorithm and analysing the differences. This error in the reconstruction was found to be of the order of $10^{-10}$.

To further explore the reconstruction in terms of wavelength and depth, multi-flash images are presented in Figure 6 below, where the features embedded in the multi-layer phantom are imaged at different imaging depths and using wavelengths from the blue to the NIR. The probes were fixed 8 mm above the sample, and images were captured at a range of imaging intensities, with the best ones for inspection being selected. A similar procedure takes place in clinical situations when examining tissue samples, where the clinician will adjust the system settings until an adequate image is generated.

The first row of images displays images under white light illumination that have not been corrected for specular reflections. Very strong surface reflections saturate the pixels and the margins of the features are no longer distinguishable. The second row of white-light-illuminated images shows the traditional guidance image (illuminated using white light) corrected for specular reflections using the multi-flash post-processing approach, and the hydroxyapatite crystal features can now be clearly distinguished. In the following rows, multi-flash corrected images are shown for the other illumination wavelenghts used. The measurement depth is varied using the multi-layer phantom by adding the corresponding layer on top of the bare tissue phantom. It is observed that as the thickness of the phantom above the feature increases the features become less discernible. The longer wavelengths can be seen to penetrate deeper into the tissue mimicking phantom and features are more clearly seen. Note that the illumination intensity is selected such that the features are the most visible to a human observer, hence the apparent decrease in brightness under the 660 nm illumination as the depth increases. Higher intensity illumination here over-saturated the image, making it impossible to discern features especially when the light is already highly scattered. Increasing wavelengths clearly allows for observation of features at greater depth. The 940 nm illumination allows features under 2.0 mm of phantom layers to be resolved, although there is some blurring observed at this depth due to scattering. In all cases, surface reflections are effectively removed from the image, revealing underlying information.

The same measurement procedure was repeated for the cross-polarised imaging probe in Figure 7. Similar to the multi-flash probe, the longer wavelengths penetrate deeper into the phantom. A comparison of the 940 nm wavelength suggests the ability to resolve features slightly better than the multi-flash probe at this NIR wavelength. Additionally, the boundaries of the features appear more well defined under all illumination wavelengths. The fibre reflections can be seen in the images captured using the cross-polarised probe and will be discussed later. These always appear at the same location.

### 3.2. Ex Vivo Human Tissue Imaging

To demonstrate the clinical value of the proposed solutions, excised human breast tissue was imaged. Figure 8 shows a comparison between a camera image with single fibre illumination, the reconstructed multi-flash image, and the cross-polarised image, respectively. It is an example of a typical clinical imaging target. Both the multi-flash reconstruction as well as the cross-polarisation effectively eliminate the specular component of the reflected light. In both cases, only the diffuse component remains providing a blemish free image that presents the information of the surface and underlying features.

The same location is imaged using white light and NIR illumination. In the white image, a set of grey blue dots can be seen in the top left quadrant of the image. Specular reflection is clearly seen in the first set of images where the sample is only illuminated without any specular reduction method using a single fibre source. Here, the glossy tissue surface is obscuring the view of the underlying morphology of the tissue. In the next two sets of images it is clear that both the multi-flash corrected image and the cross-polarised image have effectively removed the specular reflection and revealed previously obscured underlying information. Under both illumination types, white and NIR, the cross polarised image appears brighter here due to the intensity choice in the multi-flash image that returns the best result. A balance is struck between illumination intensity and output image quality, in particular in the algorithmic specular reflection removal approach. Both methods show excellent image quality and colour reproduction. This has been validated with many measurements on human tissue beyond the selection selection shown here, with further publications to follow.

## 4. Discussion

The results of cross-polarised imaging, as well as multi-flash imaging, using micro-cameras, of phantom test targets and human tissue demonstrate two effective methods for removing surface reflections. The orthogonal polariser imager rejects the polarisation maintaining direct reflections, from the glossy tissue surfaces. Imaging into the tissue by collecting only multiple-scattered light is thereby achieved. Similarly, the multi-flash imaging probe shifts the specular reflections as illumination strikes the surface from different positions. These shifts in specular reflections can be filtered out and the image below can be reconstructed. Important to note is that the two methods fundamentally interrogate different physical properteries of reflected light. The multi-flash approach effectively samples all polarisation states, whereas the polarimetry approach only samples multiple-scattered light. This should be considered when choosing a technique, while the polarisers provide a solution that is free from post processing, it is limited by the dimensional constraints of the dicing process and the specifications of the polarisers. This in turn bounds the minimum dimensions of any potential device. An alternative option may be to deposit or grow polarisers directly onto fibre tips and sensor to drive the footprint down further. For the multi-flash approach, the dimensions are bounded by the requirement to have four illuminating fibres, and the illumination cone overlap condition, as well as the illumination profile in combination with the camera. From a clinical compliance standpoint, the images from the polariser-based solution are more desirable. For real-time imaging and intraoperative scanning across tissue, the cross-polarisation probe is preferred as there is no requirement to keep the probe and tissue static during image acquisition. Currently the multi-flash approach can acquire images in approximately 0.20 s with the synchronisation of image acquisition with the optical switch being the made bottleneck. With a faster camera and a synched rapidly switching optical switch, however, these limitations could be mitigated in the future.

Illumination pattern is important so that the FOV of the camera is fully illuminated. Illuminating with four fibres simultaneously, provides a more even illumination pattern that covers a greater area of the FOV of the camera. More illumination intensity is easily provided using the illumination system. By using more powerful light sources, any intensity problems should be eliminated entirely. Fibre selection is ideally large diameter and high numerical aperture, however this can be supplemented with lens design. Important in all cases is illuminating much of the FOV as possible and therefore the cameras are selected with a low FOV to allow for a maximum illuminated area. Additionally the lower FOV micro cameras were selected due to the reduced barrel distortion observed on the graded-index lens camera cube package. This becomes more challenging when reducing footprint and trade-offs in illuminated area and illumination pattern must be made with cost. For multi-flash imaging, correct overlap of the light cones is an additional consideration so that as much area as possible is illuminated by each fibre position, ensuring that any areas with specular reflections have a shift in reflection.

Micro cameras typically provide good magnification, and the FOV in these images is of the order of 10.0 mm. This is a positive aspect, given sufficient spatial resolution and pixel count, when looking for small features in tissue. However, it can become a negative when scanning a larger region of tissue is necessary. One of the main limitations of micro camera endoscopy systems is the number of available pixels. In colour cameras, only a fraction of the pixels are active in each spectral band. This becomes more of a problem when using the pixels not just for guidance but for diagnostic measurement. Using a monochrome camera can help resolve this, as now all pixels are sensitive to a broadband visible and NIR illumination. This, in turn, increases the spatial resolution of our sensor for each of the colour bands. These colour bands can then be delivered sequentially using the aforementioned illumination system, or if desired laser illumination [36], again limited by motion of the probe or tissue in relation to acquisition time.

The ex vivo results obtained using this system demonstrates the efficacy of both specular reflection removal approaches when imaging human tissue. Figure 8 shows effective elimination of specular reflections in both modalities, with the features on the tissue surface now discernible. Figure 6 and Figure 7 present depth measurements of embedded features in multi-layer phantoms. In Figure 7, the cross polarised system fails to eliminate all of the specular reflections, as bright hot spots can be seen in almost all the images but in particular the white and NIR illuminated images. These are reflections from the illuminating fibres, which are not fully eliminated due to the limitations in the polarisation system. A number of possible effects should be considered. The first is that of the performance limitations of the polarisers leading to not all the incident and collected light being completely cross polarised. The second effect is likely an angular effect where when using a large NA camera objective, different pixels will be looking at the surface under different angles. The polarisation of the specular reflection from the surface will be parallel to the surface everywhere. Hence, depending on the location on the surface, the specular reflection reaching the camera will have a slightly different angle with the polarisation filter in the camera for each pixel. On average these align to be zero, but there will be a range of angles where transmission is non-zero. This likely reduces the overall performance of the system however in the case of the hotspots remaining in the same location the contribution is primarily that of contrast in the polarisers. The diffuse reflection is an order of magnitude lower than the specular reflection from the surface, and as a result hotpots form in certain locations. A slight angular shift in the probe positioning, i.e., the probe is not aligned perfectly perpendicular with the surface of the phantom may play an additional role. When compared with specular reflection uncorrected images, such as images in Figure 5a–d and Figure 8 however, a drastic improvement is seen. The final use case of such probes remains the clinic where conditions are never ideal. The images in Figure 8 remain indicative of the value of these techniques in particular when used on biological tissue.

Due to tissue optical properties, in the visible to NIR range, tissue can be interrogated at greater depth with increasing wavelength. From the phantom images it is clear that there is an increase in imaging depth with an increase in wavelength, as was expected. Figure 6 and Figure 7 demonstrate an ability to image into tissue at depth in particular when using NIR wavelengths. These effects are significant and this paper recommends the inclusion of NIR modalities into micro camera endoscopy systems. In many cases features of interest present themselves not just in plane with the surface. The human body often presents complicated morphology, and therefore there exists a clinical requirement to image at some depth, or detect embedded features. The removal of specular reflections improves image quality and enables real-time feature detection algorithms to detect these in the future. Flexibility in illumination wavelength is additionally beneficial for these systems, to interrogate different colour bands, and to allow the combination of wavelengths to enhance contrast. This gives clinicians tools to view a scene in the way that best suits the particular use case. Quick scanning with white light illumination allows for viewing of surface features and morphology assessment. Switching to more specific wavelengths, such as the NIR to probe at depth, when features or structured of interest present beneath the surface, or the blue to examine surface features. Wavelength flexibility also enables viewing of particular bio-markers sensitive to certain wavelengths. This can also be extended to additional imaging modalities such as diffuse reflectance spectroscopy applications. The techniques presented in this paper are adaptable to systems of all sizes, including traditional chromoendoscopy systems, and specular reflection removal improves image quality independent of system. It is of particular importance when miniaturising systems using micro cameras, where the magnification is quite high and the field of view and pixel count relatively low, and any artefacts in the image have a high impact on image quality.

This system lays the groundwork to provide clinicians with diffuse images, where acquisition is quick and simple, and the input wavelengths can be carefully controlled. A platform is presented that is adaptable to 3–7 Fr applications using the fibre illumination and micro camera. Examples where this system can provide an addition to the current workflow including in tissue-conserving surgeries to ensure that the resections margins are tumour-free, as a tool in peripheral lung applications to ensure image quality, or as a tool in imaging of vulnerable plaque in cardiovascular medicine. However, such a probe should include multiple modalities to provide an enhanced image that goes beyond simple image guidance. It would also vary in dimensions depending on the use case. Future work should therefore include adapting the probe into a multi modal system that is capable of diffuse reflectance spectroscopy as well as image guidance, with the ability to co-locate features detected with different modalities. Work is also being carried out on integrating image processing with high dynamic range and multi spectral imaging to highlight areas of interest for the clinician.

As medicine takes more and more steps towards minimally invasive approaches, it is expected that the demand for such techniques will increase. Rapid acquisition and real-time feedback for clinical use will become a standard. The techniques demonstrated in this paper are not uncommon in larger dimensions, e.g., in professional photography; however, the challenge lies in the development of small footprint systems that can find their place in a clinical environment. For clinical applications, a generalized probe for use in clinical settings has not yet been introduced. For example, imaging probes would be valuable in outward clinics in providing general practitioners or dermatologists detailed visualization of small skin abnormalities, resulting in more accurate diagnosis and treatment. Additionally, imaging probes can be extensions for surgeons in assessing tumours or lesions during surgical procedures. To optimize clinical use, direct and non-invasive assessment with real-time feedback is essential. Moreover, the probe and system should be easy to use, in order to minimize the need for technical assistance. This paper has miniaturised such a system into a 10.0 mm handheld portable footprint. The techniques presented lend themselves to being further miniaturised, with ongoing work moving towards a footprint of 2.3 mm as shown in Figure 9. At this footprint, the device will lend itself to use in the working channel of many endoscopes, for navigation deeper into peripheral lung space, for example. Camera technology is expected to continue to evolve. In the medical technology field especially the gold standard is a guidance image in situ, and so we expect further footprint reductions, and micro cameras to be used more frequently in areas previously inaccessible for imaging. For their effective use, systems must be miniaturisable to fit the small form factor the micro camera allows. Clinical relevance can also not be neglected. Portability, handheld nature, and image and information quality must all be taken into consideration. Micro cameras then open doors to miniaturising benchtop systems and providing information from areas where it was previously unavailable. Work is also ongoing to develop robust sterilizable systems suitable for use in operating theatres.

Future work will involve optimising this technology to meet clinical needs in particular clinical applications, providing valuable information to clinicians which was previously unavailable. This includes further miniaturisation if necessary, in combination with image processing and feature recognition software to improve output images, and inclusion of diffuse reflectance imaging modes. The current system is built to support these modalities. Consideration should also be given to improving image quality by using monochrome cameras instead of RGB cameras, and further exploration of sequential illumination is necessary. Most importantly, all of this work should be performed with the clinical requirements in mind. This means in close collaboration with clinicians, with the systems being used in a clinical environment.

## 5. Conclusions

Surface reflections are often undesirable and make it difficult to determine with certainty whether a feature is present in the image or not, complicating the determination of sub-surface tissue structures. This can cause erroneous results in human and machine analysis, and is not desirable in clinical care, where quick representation of important data is critical. In this paper, two handheld systems of the same footprint, capable of producing specular-reflection-reduced images of tissue using two different techniques, have been presented. These probes use micro-camera technology, which provides a flexible imaging platform and represents an emerging technology in surgical settings. The design is handheld and portable to easily fit into clinical workflow, as well as miniaturisable. To remove undesirable specular reflections two techniques are implemented. One, the integration of cross-polarisation using diced high-quality polarisers, and two, multi-flash imaging to shift reflections and filter them out algorithmically, in a post processing step. The use of these techniques enables the generation of images that are free from undesirable surface reflections. Both techniques achieve the goal of removing surface reflections on various targets including resected breast specimens, returning more suitable images for visual inspection. Advantages and disadvantages of both methods are discussed. Phantom verification shows the ability to image features at a depth of up to 2.0 mm with both techniques when using NIR illumination. The polarisers show clearer edges of the features upon visual inspection, although require more illumination power due to their attenuation. Both techniques are further miniaturisable, the multi-flash technique requiring four illumination sources to be integrated, bounding the minimum dimensions, and these sources need to be switched on sequentially. The multi-flash approach therefore also requires the camera probe to be stationary for a fraction of a second, whereas the polarised probe is a solution that works independent of motion. The miniaturisation of the cross-polarised approach is dimensionally bounded by the size of the polarisation. In the clinical environment, the cross-polarised approach is currently preferred. In conclusion, two approaches to specular-reflection removal for endoscopy are presented. A balance must be struck between miniaturisation, acquisition time, interaction with other system modalities, and application, while design decisions for endoscopic applications are application-specific to a high degree. It is shown that specular reflection problems must be considered, and flexibility in illumination wavelengths including in the NIR is recommended. Both options presented show promise for use in a micro-camera platform for examining tissue.

## Figures and Tables

**Figure 1 micromachines-14-01062-f001:**
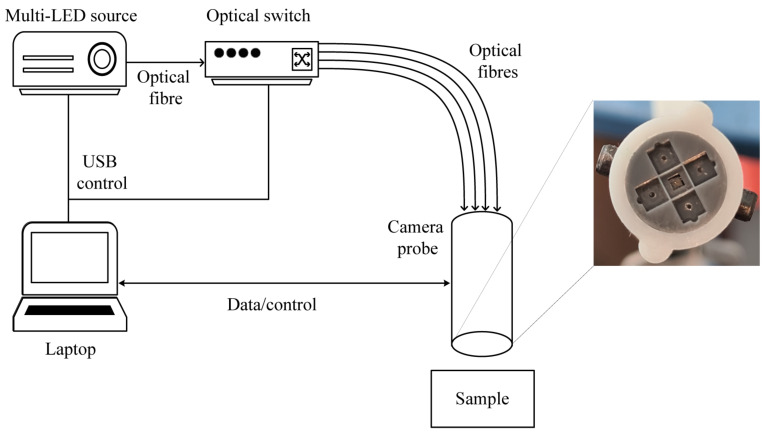
Schematic diagram of the experimental arrangement for the multi-flash measurements and close-up view of the multi-flash probe showing central camera sensor and concentrically arranged illumination fibres.

**Figure 2 micromachines-14-01062-f002:**
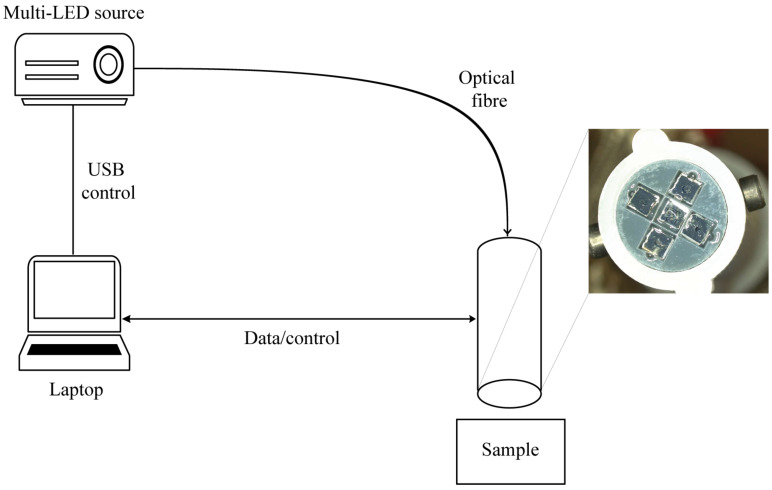
Schematic diagram of the experimental arrangement for the cross-polarisation measurement and close-up view of the polariser probe showing central camera sensor and concentrically arranged illumination fibres, each with a small diced glass polariser mounted in front. The polarisers on the camera and image fibres, respectively, are positioned orthogonally to one another.

**Figure 3 micromachines-14-01062-f003:**
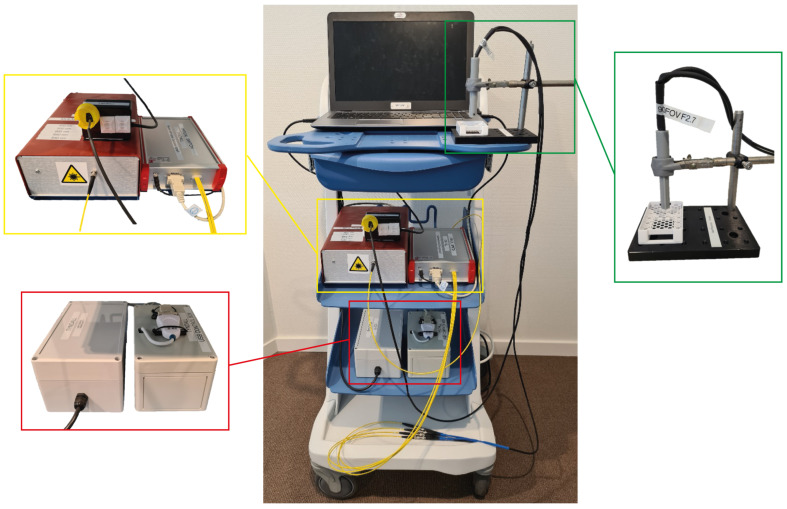
Entire clinical trolley with light source and optical switch (yellow), power supply and data acquisition box (red), and camera probe with sample holder (green).

**Figure 4 micromachines-14-01062-f004:**
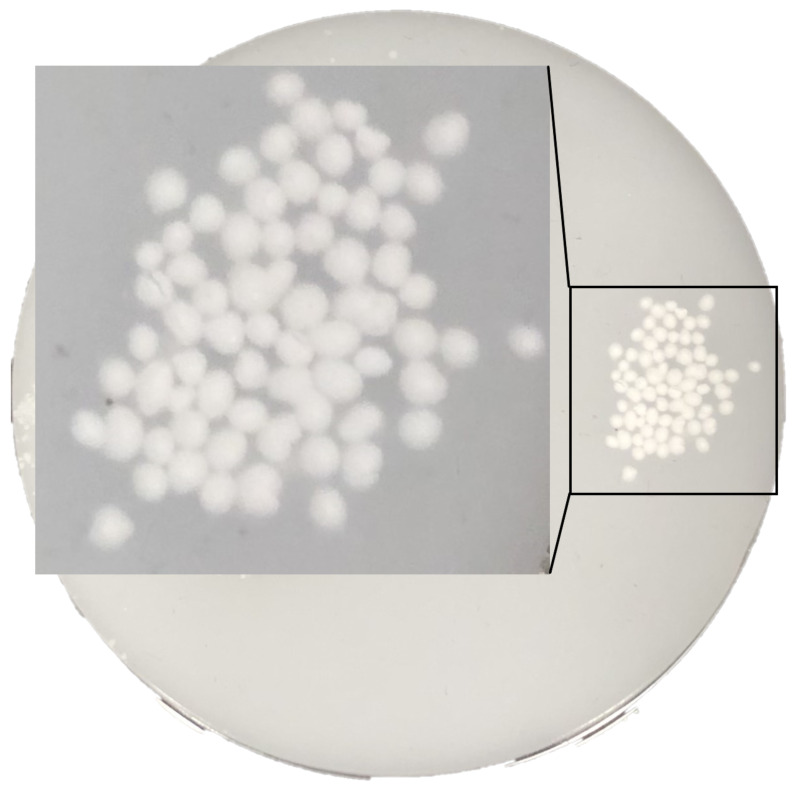
Image of base layer of the tissue-mimicking phantom developed for this study, with a close-up view of the 1.0 mm features imaged in this work.

**Figure 5 micromachines-14-01062-f005:**
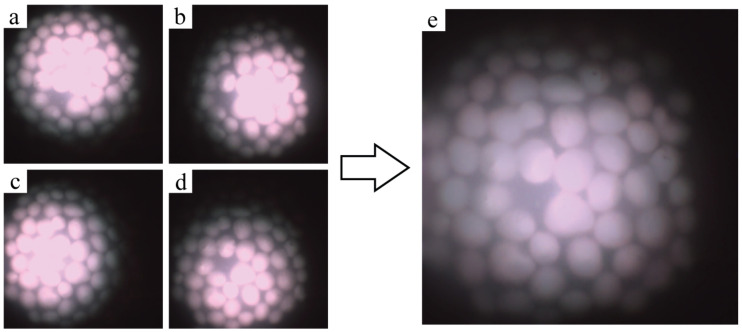
Images displaying four separately illuminated white light images of the same site on the phantom from different illumination locations (**a**–**d**) and the reconstructed output of the multi-flash specular reflection reduction algorithm (**e**).

**Figure 6 micromachines-14-01062-f006:**
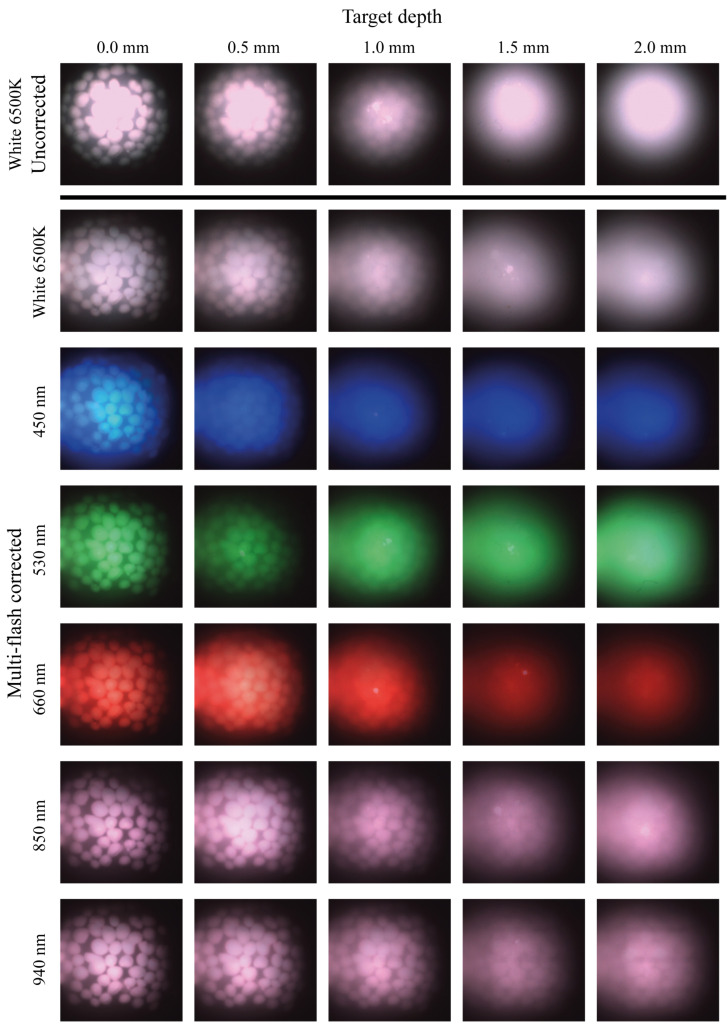
Uncorrected images of tissue-mimicking phantoms with embedded features shown under white illumination in the first row. Reconstructed multi-flash images of tissue-mimicking phantoms at different imaging depths and different illumination wavelengths shown below.

**Figure 7 micromachines-14-01062-f007:**
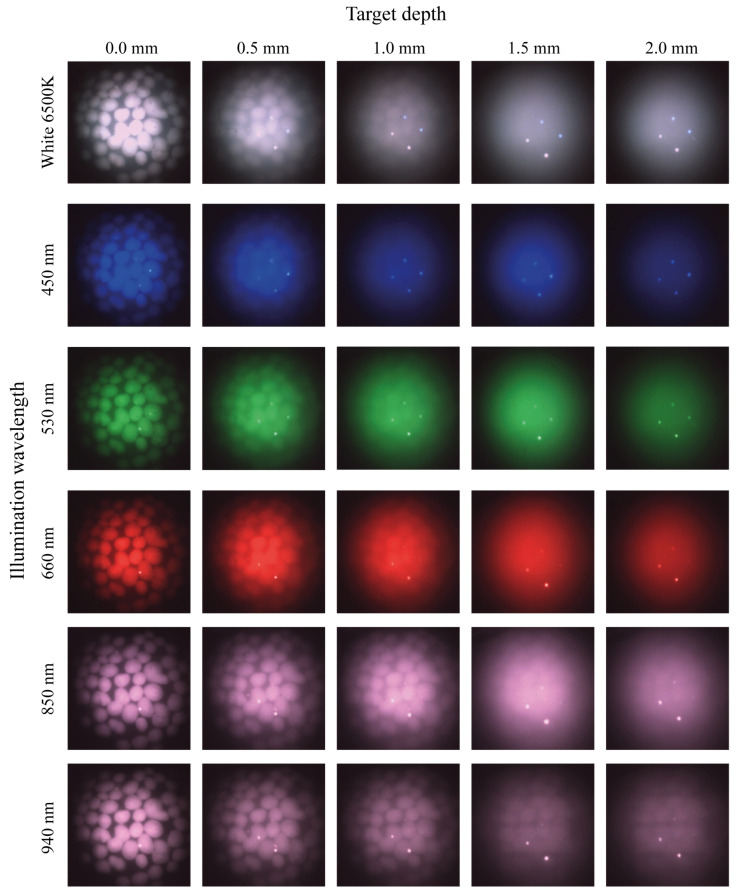
Cross-polarised images of tissue-mimicking phantoms at different imaging depths and different illumination wavelengths. Note the four bright spots that are visible on most of the images are direct reflections from the four illumination fibres. The surface of the phantom is very flat and reflective and the polarisers fail to filter out these brightest spots.

**Figure 8 micromachines-14-01062-f008:**
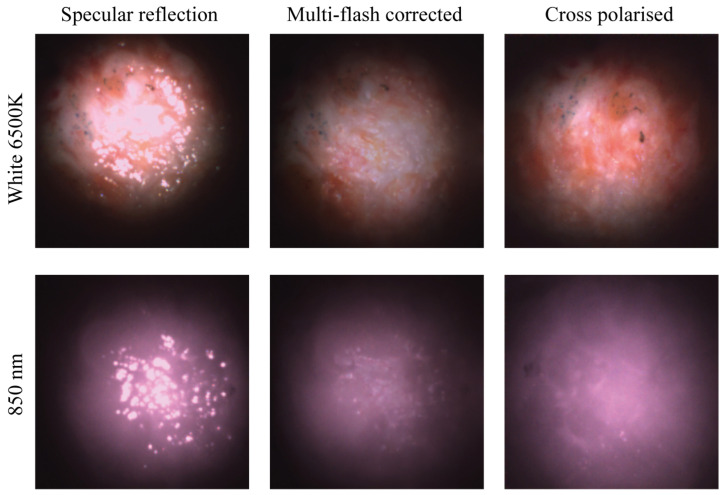
Humanbreast tissue images showing single fibre illumination with no polarisation (**Left**), reconstructed image after multi-flash post processing (**Middle**), and cross-polarisation imaging (**Right**).

**Figure 9 micromachines-14-01062-f009:**
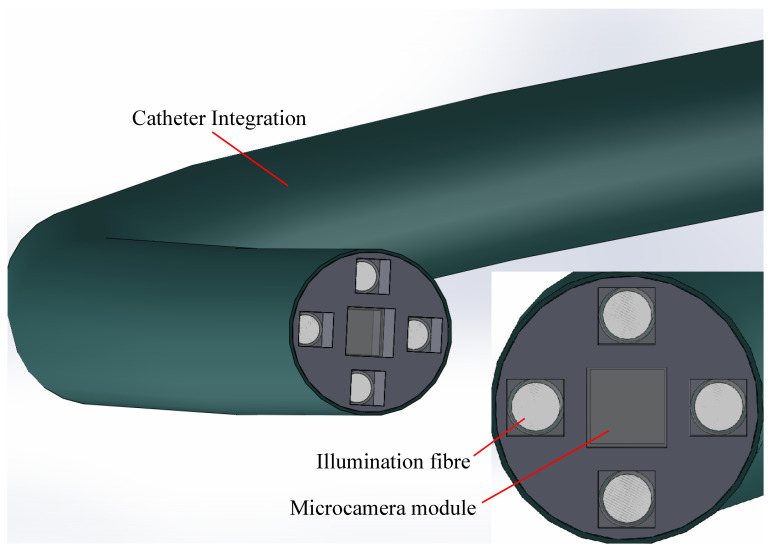
Schematic representation of the miniaturisation possibilities for the two implemented specular-reflection-reduction methods. In this case, a smaller camera sensor OV6948 will be utilised with 0.50 mm and 0.70 mm polarisers integrated in front of the sensor and the 0.40 mm illumination fibres.

## Data Availability

Data underlying the phantom results presented in this paper are being made publicly available, and will be accessible at DOI:10.5281/zenodo.7709437. Tissue images are available from the authors upon reasonable request.

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
