# Peer review of "Microcamera Visualisation System to Overcome Specular Reflections for Tissue Imaging"

_micromachines, 2023, doi:10.3390/mi14051062_

Round 1

Reviewer 1 Report

The article titled "Microcamera Endoscopy System to Overcome Specular

Reflections for in vivo Tissue Imaging" reports the development and testing of two different miniature camera probes designed to improve the quality of medical imaging by reducing specular reflections from glossy tissue surfaces.

The authors present two modalities for reducing specular reflections: (1) the multi-flash technique and (2) the cross-polarization technique.

Validation experiments are limited to tissue-mimicking phantoms and excised human breast tissue. They report reduced distortion and artifacts caused by specular reflections and provide clear, detailed images of tissue structures.

Overall, the article is well-written and of interest to this community. The article may be improved by considering the following comments:

1.     I would suggest the authors omit the In-vivo imaging phrase in the title since the current study does not support the statement and only reports feasibility in-silico and ex-vivo.

2.     Quantitative evaluation of imaging is lacking in the current manuscript. The authors must present a study on how much improvement was achieved by correcting uncorrected images.

3.     Abstract and Conclusion should report quantitative evaluation information not present here.

Therefore, the manuscript's present form will benefit from a minor revision.

Reviewer 2 Report

The paper aims to advance techniques to remove specula reflection by a small camera, which can be assembled on the tip of optical fiber. However, the paper is not well written, especially not clear to the reader who are not much in the field of cameras and image processing. Since this journal is with a focus on micromachines, it is reasonable to assume that the reader may not have much knowledge behind the image processing, and image science. My specific comments are as follows:

1.      “Two small form factor camera probes ...” Comment: please explain what is called small factor in the context?

2.      Is the contribution about small camera or miniaturization of camera or about the algorithm to remove the specula reflection?

3.      The situation, where the camera stays, needs to be clearly described.

4.      The contribution of miniaturization systems is not clear.

5.      In terms of removal of specular reflections, not clear about the contribution: principle, algorithm, or system?

Reviewer 3 Report

From my point of view, the work presented by the authors is a first approximation for the possible future clinical use of an image enhancement system. They have used a phantom model that allows defining improvements in image quality by filtering light reflections. However, the phantom is still quite far from a real human organ and even more so from the live observation of a human or animal organic segment. The authors state that this system is being developed with the aim of being able to apply it in future endoscopic examinations. In this sense, it would probably be useful if the system or equipment is friendly to be integrated into an endoscopic system and for this it is necessary to consider the space required for its integration into an endoscope. Therefore, they should discuss its ability to be merged with another team and the space it would occupy within that team. From the point of view of endoscopies that generally observe mucosa in vivo, the breast tissue model is far from being close to the observation of the mucosa. In the real life, the mucous tissue have liquid and secretions on their surface that further increase the reflection and alter the brightness, so their use should be simulated in conditions closer to that. Likewise, it would be desirable for the authors to comment on whether the system would be friendly and if it would be a possible contribution to the image quality currently provided by endoscopic systems with high-resolution cameras coupled to electronic chromoendoscopy systems.

Round 2

Reviewer 2 Report

I am generally satisfied with the revision along with rebuttal.